# Visual Para-Thinker: Divide-and-Conquer Reasoning for Visual Comprehension

Haoran Xu [*1]   Hongyu Wang [*2]   Jiaze Li [†‡3]   Shunpeng Chen [3]   Zizhao Tong [4]   Jianzhong Ju [3]   Zhenbo Luo [3]   Jian Luan [3]

## Abstract

Existing LLM test-time scaling laws emphasize the emergence of self-reflective behaviors through extended reasoning length. Nevertheless, this vertical scaling strategy often encounters plateaus in exploration as the model becomes locked into specific thinking pattern. By shifting from depth to parallelism, parallel thinking mitigates the narrowing of exploration. However, the extension of this paradigm to visual domain remains an open research question. In this paper, we first examine the role of visual partitioning in parallelized reasoning and subsequently propose two distinct strategies. Based on the above, we introduce Visual Para-Thinker, representing the inaugural parallel reasoning framework for MLLMs. To maintain path independence and promote diversity in reasoning, our approach integrates Pa-Attention alongside LPRoPE. Leveraging the vLLM framework, we have developed a native multimodal implementation that facilitates high-efficiency parallel processing. Empirical results on benchmark datasets such as V*, CountBench, RefCOCO, and HallusionBench confirm that Visual Para-Thinker successfully extends the benefits of parallel reasoning to the visual domain.

## 1. Introduction

The remarkable success of Large Language Models (LLMs) is intrinsically linked to their ability to reason through complex problems. Existing research predominantly focuses on bolstering the reasoning capabilities of LLMs by increasing the depth and length of their reasoning trajectories. The Chain-of-Thought (CoT) (Wei et al., 2023; Yao et al., 2023) prompting mechanism introduces explicit, meticulously de-signed, and sequential reasoning steps to guide the model toward the final answer. Furthermore, Reinforcement Learning with Verifiable Rewards (RLVR) (Guo et al., 2025; Team et al., 2025), which improves model accuracy by incentivizing unconstrained exploration. This approach facilitates the emergence of self-reflective capabilities within the model. Notably, both GPT-5 (Singh et al., 2025) and DeepSeek-V3.2 exemplify frontier, large-scale LLMs that leverage such complex, multi-step reasoning schema.

Contemporary research (Wen et al., 2025; Zheng et al., 2025b; Wu et al., 2025; Huang et al., 2026a; Team et al., 2026a) has begun transitioning toward a parallel thinking framework for test-time scaling, as illustrated in Fig 2(b). The efficacy of this pattern was recently demonstrated by Google's Gemini (Pichai et al., 2025), which leveraged parallel thinking to excel at the International Mathematical Olympiad. Similarly, LongCat-Flash-Thinking (Team et al., 2026b) introduced the parallel thinking mode. In contrast to sequential reasoning, which focuses on extending the depth of the inferential chain, parallel reasoning expands the search space along the dimension of width by generating multiple concurrent reasoning paths. This paradigm shift addresses the inherent limitations of conventional vertical expansion, which frequently suffers from diminishing exploratory returns as models become confined to rigid, singular inferential trajectories—thereby stifling the discovery of diverse problem-solving strategies.

Despite this scaling paradigm success in text-centric tasks like mathematics and coding, applying this paradigm to visual reasoning remains a critical research lacuna due to the unique structural challenges inherent in the vision domain.

Leveraging the parallel reasoning paradigm, we first investigate the impact of visual partitioning in the visual domain and introduce two distinct strategies: Block-based partitioning and Scan-order partitioning. Building upon these insights, we introduce Visual Para-Thinker, the first framework for parallel multimodal reasoning in Large Multimodal Models (LMMs). To ensure the reasoning path isolation, unbiasedness, and discriminability, we propose Path-aware Attention (**Pa-Attention**) and Learnable Parallel Rotary Position Embedding (**LPRoPE**). The latter integrates the conventional RoPE (Su et al., 2023) scheme with learnable

---

[*]Equal contribution ,[†]Project Leader,[‡]Corresponding Author [1]Zhejiang University [2]Hunan University [3]MiLMPlus, Xiaomi Inc [4]University of Chinese Academy of Sciences. Correspondence to: Jiaze Li <ljzazsl@gmail.com>.

*Proceedings of the 43rd International Conference on Machine Learning*, Seoul, South Korea. PMLR 306, 2026. Copyright 2026 by the author(s).

path-specific position encodings to effectively distinguish between parallel reasoning trajectories.

Finally, in order to ensure the efficiency and parallelism of Visual Para-Thinker, our implementation leverages and extends the vLLM (Kwon et al., 2023) framework through a series of engineering enhancements. To demonstrate the effectiveness of our approach, we conducted extensive experiments across various tasks, including counting, visual grounding, fine-grained perception, and hallucination. The results consistently validate the efficacy of our proposed method. Our core contributions are summarized as follows:

- We investigate the impact of visual partitioning and introduced two distinct strategies.

- We propose Visual Para-Thinker, the first parallel thinking framework designed for MLLMs.

- We integrate the Pa-Attention and LPRoPE mechanism into our method, enabling the path isolation, unbiasedness, and discriminability.

- To maintain path parallelism and efficiency, we implemented engineering enhancements leveraging the vLLM framework.

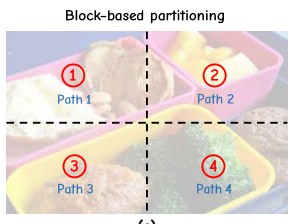 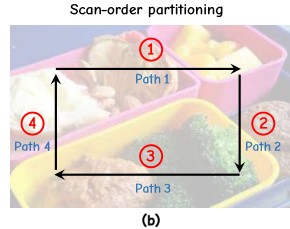

*Figure 1.* Schematic representations of two distinct strategies for visual partitioning. (a) illustrates Block-based partitioning, while (b) shows Scan-order partitioning.

## 2. Motivation

While previous studies have established the efficacy of parallel reasoning in text-based tasks, this section elucidates visual partitioning within the Visual Para-Thinker framework. We place particular emphasis on ensuring the diversity of reasoning paths centered on visual information.

**Theorem 2.1.** *The parallel reasoning paradigm necessitates diversity across various reasoning paths. In this context, the diversity of visual paths is fundamentally rooted in the distinct patterns of visual attention distribution.*

Recent studies (Xiao et al., 2024; Xu et al., 2025) have demonstrated that the attention sink phenomenon significantly influences model performance. Building upon the premise that parallel thinking necessitates distinct patterns for different reasoning trajectories, and drawing on the insight that the visual attention sink represents a specific attention allocation pattern (Kang et al., 2025), we posit that parallel reasoning paths are essentially manifestations of diverse visual attention allocation strategies. Based on the above insights, we propose two distinct strategies for visual partitioning: Block-based partitioning and Scan-order partitioning. Fig 2(a) illustrates the attention allocation for Path 1 and Path 4 under the Block-based partitioning strategy. While Path 1 exhibits a higher concentration of attention on the top-left block, Path 4 primarily focuses on the bottom-right region. This divergence in attention distribution results in the emergence of distinct reasoning trajectories within visual partitioning.

**Block-based partitioning.** This strategy partitions reasoning paths based on specific regional subgraphs. In this configuration, each path captures a unique visual attention distribution centered on a designated sub-region, such as the top-left, top-right, bottom-left, or bottom-right quadrants, as illustrated in Fig 1(a).

**Scan-order partitioning.** This approach differentiates reasoning paths by employing various visual scanning trajectories. Specifically, each path represents a distinct allocation of visual attention that corresponds to a predefined scanning order, such as left-to-right, top-to-bottom, right-to-left, and bottom-to-top, as depicted in Fig 1(b).

Our method employs a hybrid training scheme that integrates both Block-based and Scan-order partitioning.

## 3. Method

### 3.1. Preliminaries

RoPE (Su et al., 2023) is a mainstream approach for encoding positional information in modern LLMs. It applies a rotational transformation to the query and key vectors, thereby incorporating relative position dependencies directly into the self-attention mechanism. Specifically, Given a query vector $\mathbf{q}_m$ at position $m$ and a key vector $\mathbf{k}_n$ at position $n$, the attention scores $\mathbf{S}_{m,n}$ are calculated as:

$$\mathbf{S}_{m,n} = (\mathbf{R}_m \mathbf{q}_m)^T (\mathbf{R}_n \mathbf{k}_n) = \mathbf{q}_m^T \mathbf{R}_{n-m} \mathbf{k}_n \quad (1)$$

$\mathbf{S}_{m,n}$ merely depends on m, n and their relative positions. In addition, $\mathbf{R}_m$ is the rotation related to the position $m$ and the frequency $\theta_i = \beta^{\frac{-2i}{d}} (i = 0, \ldots, \frac{d}{2} - 1)$.

### 3.2. Model Design

In this section, we present the design scheme of Visual Para-Thinker as Fig 3. To be specific, visual para-thinker is divided into two stages: Parallel Reasoning and Summary.

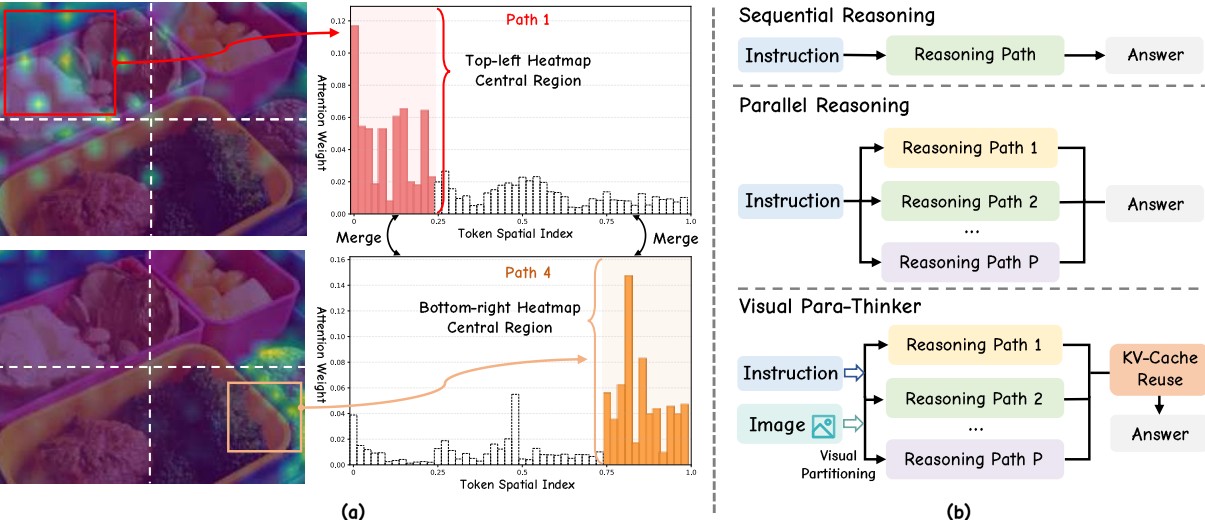

*Figure 2.* (a) illustrates the attention allocation results for Path 1 and Path 4 using the Block-based partitioning strategy during visual partitioning. The left panels present the attention maps for path 1 and path 4, while the right panels display the corresponding histograms of the spatial attention weight distributions. (b) illustrates a comparison between various test-time scaling paradigms.

- **Parallel Reasoning**: Based on the shared context, the thinking directions of different reasoning paths are allocated using the idea of visual partitioning.

- **Summary**: Merge the contexts of different parallel reasoning paths and comprehensively consider them to reach the final conclusion.

Guided by the four foundational principles of parallel thinking—path isolation, unbiasedness, discriminability, and parallelism—we introduce our framework, Visual Para-Thinker, comprising Pa-Attention, LPRoPE, and Framework Implementation. Specifically, Pa-Attention is designed to enforce structural path isolation, whereas LPRoPE ensures both unbiasedness and discriminability across paths. Finally, our implementation enables high-efficiency path parallelism.

### 3.2.1. PA-ATTENTION

**Definition 3.1** (Reasoning Path Isolation). The property of Reasoning Path Isolation stipulates that the computational trajectories of distinct reasoning paths remain mutually independent. Formally, the hidden states and output tokens of a given path $r^{(i)}$ must be invariant to the existence or content of any parallel path $r^{(j)}$.

**Theorem 3.2.** *Pa-Attention has reasoning path isolation.*

The vanilla causal attention mechanism is unable to isolate different paths, and information within different paths will leak out. Therefore, in order to ensure the Reasoning Path Isolation, we design Pa-Attention as Fig 3. Specifically, in the parallel reasoning stage, Pa-Attention treats each path

as a window and restricts each token within a `<think i>` block to attend only to tokens from the same path and the shared context. In the summary stage, based on the shared context set $P$, the summary set $S$ is generated by integrating multiple reasoning paths $r^{(i)}$. Let $M_{i,j}$ denote the attention mask between the token index $i$ and index $j$. The mathematical expression is as follows:

$$M_{i,j} = \mathbb{1}\Big(j \leq i\Big) \cdot \mathbb{1}\Big(\text{is\_visible}(i,j)\Big) \qquad (2)$$

$$\text{is\_visible}(i,j) = \begin{cases} 1 & \text{if } j \in P \vee i \in S \\ 1 & \text{if } \exists k : \{i,j\} \subseteq r^{(k)} \\ 0 & \text{otherwise} \end{cases} \qquad (3)$$

### 3.2.2. LPRoPE

In the summary stage, merging multiple reasoning paths poses challenges due to the positional bias. If the different parallel reasoning paths have the same position encoding, then MLLMs will confuse the different reasoning paths, ultimately leading to confusion in reasoning. Based on the above insights, Multiverse (Yang et al., 2025a) position encodings assign a disjoint set of position indices to each path, ensuring that the positional embedding space does not overlap. However, we believe that this position id allocation method will result in a clear sequence between different paths, which in turn leads to preconceived positional biases among different paths during the summary stage. Therefore, in order to ensure the reasoning path discriminability and reasoning path unbiasedness, we integrate LPRoPE into Visual Para-Thinker.

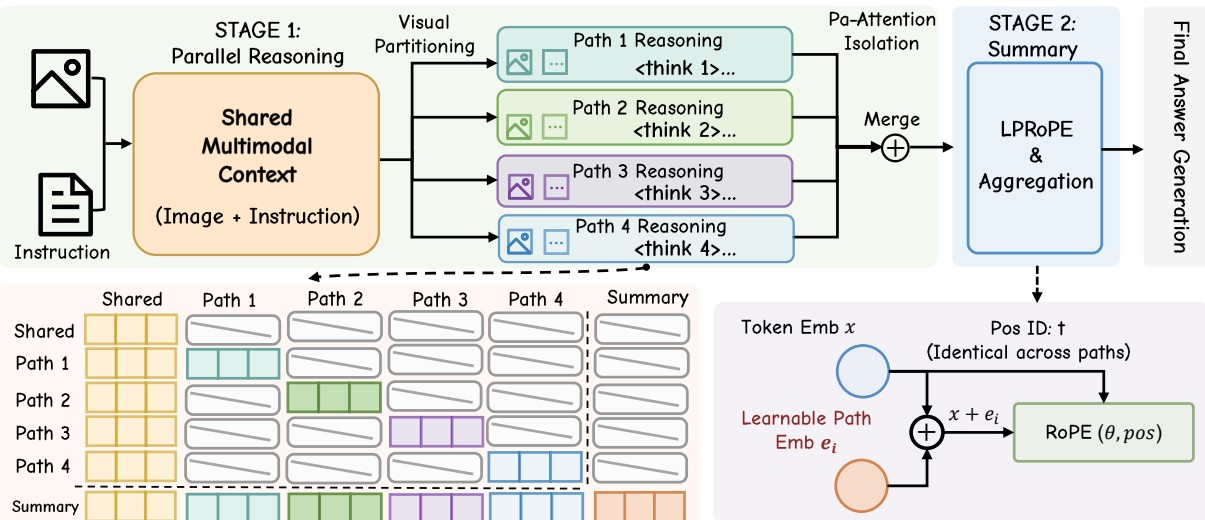

*Figure 3.* Visual Para-Thinker architecture. Our framework operates in two stages, namely Parallel Reasoning stage and Summary stage. In the Parallel Reasoning stage, multiple reasoning paths are generated through visual partitioning. These reasoning paths are isolated via Pa-Attention and identifiable through LPRoPE. Subsequently, in the Summary stage, the contexts from these isolated reasoning paths are integrated to derive the final output.

**Definition 3.3** (Reasoning Path Unbiasedness). It represents the model's ability to maintain a uniform prior across diverse reasoning paths, avoiding exhibiting an a priori preference for any given reasoning trajectory.

**Definition 3.4** (Position ID Uniformity). In the context of multi-path reasoning, it refers to a configuration where tokens at the same relative depth across distinct reasoning trajectories are assigned identical position indices.

**Theorem 3.5.** *LPRoPE has reasoning path unbiasedness.*

Specifically, in the parallel reasoning stage, our method generates a set of $n$ reasoning paths $\{r^{(1)}, r^{(2)}, \dots, r^{(n)}\}$ for a shared input prompt $x$. The starting position ids of these reasoning paths are the same as shown below:

$$Pos_{r^{(1)}}^{start} = Pos_{r^{(2)}}^{start} \dots = Pos_{r^{(n)}}^{start} \qquad (4)$$

where $Pos_{r^{(i)}}^{start}$ represents the position id of the first token of the $i$-th reasoning path. Then the position id of each path is incremented independently. Notably, regardless of path length, the position IDs at corresponding indices are identical across all reasoning paths.

Moreover, in the summary stage, we obtain the context of all the reasoning paths. During the summary stage, the position id is incremented starting from the longest reasoning path as follows:

$$Pos_S^{start} = max(Pos_{r^{(1)}}^{end}, \dots, Pos_{r^{(n)}}^{end}) + 1 \qquad (5)$$

where $Pos_S^{start}$ is the position id of the starting token for the summary stage.

Therefore, the position embeddings of different reasoning paths are the same and LPRoPE possesses reasoning path unbiasedness.

**Definition 3.6** (Reasoning Path Discriminability). It is defined as the model's ability to distinguish and decouple tokens from various reasoning paths, ensuring that the semantic and logical identity of each path is preserved.

**Theorem 3.7.** *LPRoPE has reasoning path discriminability.*

Specifically, we add the learnable path embedding $e_i$ to the key and value embeddings of all tokens within the $i$-th reasoning path, which distinguishes each reasoning path in the summary stage. Notably, the learnable path embedding is needed to add to the key before the RoPE rotation is applied. Let $\mathbf{k}_m^{(i)}, \mathbf{v}_m^{(i)}$ denote the key and value for token $m$ at reasoning path $i$, respectively. LPRoPE is mathematically represented as follows:

$$\mathbf{k}_m^{(i)} = \mathbf{R}_m\big(\mathbf{k}_m^{(i)} + e_i\big) \qquad (6)$$

$$\mathbf{v}_m^{(i)} = \mathbf{v}_m^{(i)} + e_i \qquad (7)$$

Therefore, we enable different parallel reasoning paths to be distinguishable. Based on the RoPE formula, we can obtain the following:

$$\begin{aligned}
\mathbf{S}_{m,n} &= (\mathbf{R}_n\mathbf{q}_n^{(i)})^T\mathbf{k}_m^{(i)} \\
&= (\mathbf{R}_n\mathbf{q}_n^{(i)})^T\left[\mathbf{R}_m\left(\mathbf{k}_m^{(i)} + e_i\right)\right] \\
&= (\mathbf{q}_n^{(i)})^T\mathbf{R}_n^T\mathbf{R}_m\mathbf{k}_m^{(i)} + (\mathbf{q}_n^{(i)})^T\mathbf{R}_n^T\mathbf{R}_m e_i \quad (8)
\end{aligned}$$

Utilizing the property of rotary matrices where $\mathbf{R}_n^T\mathbf{R}_m =$

$\mathbf{R}_{m-n}$, we obtain:

$$\mathbf{S}_{m,n} = (\mathbf{q}_n^{(i)})^T \mathbf{R}_{m-n} \mathbf{k}_m^{(i)} + (\mathbf{q}_n^{(i)})^T \mathbf{R}_{m-n} e_i \quad (9)$$

- **Case 1:** When token indices $m$ and $n$ reside within the same reasoning path, LPRoPE approximates the behavior of standard RoPE, where the attention score is predominantly determined by the relative position $m - n$.

- **Case 2:** When token indices $m$ and $n$ belong to distinct reasoning paths, LPRoPE depends not only on the relative positions within the respective paths but also on the path-aware embeddings $\mathbf{e}_i$.

In both scenarios, LPRoPE exhibits the capability to effectively distinguish and identify different reasoning paths.

### 3.2.3. FRAMEWORK IMPLEMENTATION

**Definition 3.8** (Reasoning Path Parallelism). It dictates that different reasoning paths should be decoded in parallel rather than sequentially during the inference phase.

Our method is constructed based on the vLLM framework. Specifically, our implementation utilizes vLLM to parallelize reasoning path, achieving highly efficient engine-level inference. In addition to reasoning path parallelism, our framework also implements the management of the KV cache and the reasoning path controllability. Fig 4 illustrates the schematic diagram of our proposed framework implementation.

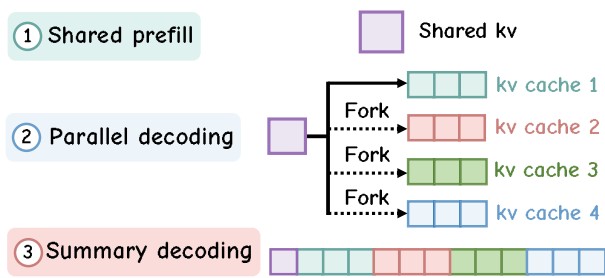

*Figure 4.* Inference framework scheme of Visual Para-Thinker. Our inference framework is divided into three stages: Shared prefill, Parallel decoding, and Summary decoding. Shared prefill generates a common KV cache, while parallel decoding produces path-specific caches that are subsequently integrated during summary decoding.

**Reasoning path parallelism**: During the parallel reasoning stage, different paths need to be executed in parallel. Specifically, during the decoding stage, we reuse the beam search decoding method and achieve parallel decoding of shared context through the fork sequence operation. In this

way, we can achieve path isolation in the inference engine without having to modify the attention mask.

**KV cache management**: Different parallel paths generate different KV caches, that is $\{cache_1, \ldots, cache_n\}$. Among them, $cache_i$ represents the KV cache generated by the $i$-th reasoning path. During the summary stage, we merge the KV caches of different reasoning paths $\{cache_1, \ldots, cache_n\}$ and the KV caches that share the context $cache'$. This implementation can save the copy of the KV cache through page attention, thereby reducing the prefill overhead in the summarization stage.

**Reasoning path controllability**: In order to maintain the controllability of the parallel reasoning mode, we have set up the parallel reasoning mode and the non-parallel reasoning mode. In parallel reasoning mode, the framework triggers specific reasoning paths by forcing the output of the special token `<think i>`. To streamline training, the special token `<think i>` is masked and serves solely as a structural guide; the model relies on LPRoPE to distinguish between reasoning paths.

## 4. Experiment

In this section, we design various experiments to prove the effectiveness and efficiency of our method.

### 4.1. Experimental Setup

**Majority voting Training Details:** Our experiments are based on a Qwen-2.5-VL-3B-Instruct(Bai et al., 2025b) and Qwen-2.5-VL-7B-Instruct model, which we denote as original models below. During training, we adopted the Pa-Attention mechanism that we proposed. Moreover, the learning rate is set to $1 \times 10^{-5}$, with a batch size of 1, the gradient accumulation steps of 8, and the maximum context length of 32k tokens. Within this 32K context length, the minimum number of visual tokens is 4 and the maximum is 16384. The model is trained for 2 epochs.

**Data Construction.** For the training recipe, we construct a parallel reasoning dataset with 163k question-answer pairs, sourced from LVIS(Gupta et al., 2019), LAION(Schuhmann et al., 2022), Microsoft COCO(Lin et al., 2015), PixMo-Count(Deitke et al., 2024), RefCOCO(Kazemzadeh et al., 2014), RefCOCO+, and RefCOCOg(Yu et al., 2016). For our data construction framework, Qwen3-VL-235B-A22B-Instruct(Bai et al., 2025a) serves as the teacher model. We generate four visual-centric reasoning paths per sample by implementing a hybrid visual partitioning strategy that integrates both Block-based partitioning and Scan-order partitioning at the temperature of 0.1. In addition, we also utilized the high-temperature Qwen3-VL-30B-A3B-Instruct and InternVL3_5-241B-A28B(Wang et al., 2025a) to generate more diverse data and check samples.

*Table 1.* Comprehensive evaluation of models on vision-centric task, including counting, fine-grained perception, hallucination task. **Bold** text indicates the best performance. Majority voting@4 means generating four times.

| Model | Size | Counting | | | Avg. | Fine-grained Perception | Hallucination | | Avg. |
|---|---|---|---|---|---|---|---|---|---|
| | | Pixmo | | CountBench | | V* | MMVP | HallusionBench | |
| | | val | test | test | | | - | - | |
| *Frontier Models* | | | | | | | | | |
| GPT5-mini(Singh et al., 2025) | - | - | 54.7 | 84.4 | 69.5 | 78.6 | 65.3 | 63.2 | 64.2 |
| GPT-4o(Hurst et al., 2024) | - | - | 54.4 | 85.7 | 70.5 | 73.9 | 63.8 | 61.4 | 62.6 |
| Gemini2.5-Pro(Comanici et al., 2025) | - | - | 59.8 | 89.7 | 74.7 | 76.7 | 69.8 | 63.7 | 66.7 |
| Claude-4-Sonnet(Anthropic, 2025) | - | - | 53.5 | 90.2 | 71.9 | 45.0 | 63.9 | 59.2 | 61.5 |
| Qwen2.5-VL(Bai et al., 2025b) | 72B | 74.0 | 70.4 | 86.7 | 77.0 | 76.9 | 72.1 | 70.0 | 71.1 |
| *Peer models* | | | | | | | | | |
| Qwen2.5-VL | 3B | 50.1 | 51.6 | 69.2 | 57.0 | 63.3 | 59.2 | 56.1 | 57.7 |
| Qwen2.5-VL | 7B | 63.0 | 67.7 | 83.1 | 71.3 | 75.9 | 68.3 | 66.0 | 67.2 |
| Sequential(SFT) | 3B | 51.5 | 52.1 | 70.4 | 58.0 | 65.0 | 60.0 | 58.3 | 59.2 |
| Sequential(SFT) | 7B | 65.5 | 68.0 | 83.7 | 72.4 | 76.1 | 68.7 | 66.7 | 67.7 |
| GRPO | 3B | 53.0 | 52.9 | 71.2 | 59.0 | 70.4 | 62.1 | 60.4 | 64.3 |
| GRPO | 7B | 64.4 | 69.2 | 84.2 | 72.6 | 78.4 | 70.9 | 68.8 | 72.7 |
| Majority voting@4 | 3B | 52.1 | 52.8 | 69.8 | 58.2 | 64.7 | 60.1 | 58.1 | 59.1 |
| Majority voting@4 | 7B | 63.2 | 67.6 | 83.2 | 71.3 | 75.6 | 68.9 | 66.8 | 67.8 |
| Visual Para-Thinker | 3B | **55.0** | **54.4** | **72.6** | **60.7** | **75.9** | **63.6** | **62.2** | **67.2** |
| Δ *(vs Qwen2.5-VL)* | 3B | *+4.9* | *+2.8* | *+3.4* | *+3.7* | *+12.6* | *+4.4* | *+6.1* | *+9.5* |
| Visual Para-Thinker | 7B | **65.7** | **70.8** | **85.4** | **74.0** | **82.2** | **71.3** | **71.0** | **74.8** |
| Δ *(vs Qwen2.5-VL)* | 7B | *+2.7* | *+3.1* | *+2.3* | *+2.7* | *+6.3* | *+3.0* | *+5.0* | *+7.6* |

*Table 2.* Performance comparison on grounding tasks including The reporting metric is Acc@0.5 (%). **Bold** text indicates the best performance.

| Datasets | Qwen2.5-VL-3B | Sequential | Visual Para-Thinker-3B |
|---|---|---|---|
| RefCOCO$_{val}$ | 85.4 | 85.7 | **88.1** |
| RefCOCO$_{testA}$ | 86.1 | 87.3 | **89.8** |
| RefCOCO$_{testB}$ | 83.0 | 84.2 | **87.7** |
| Avg | 84.8 | 85.7 | **88.5** |
| RefCOCO+$_{val}$ | 76.5 | 79.2 | **83.5** |
| RefCOCO+$_{testA}$ | 82.4 | 84.0 | **87.2** |
| RefCOCO+$_{testB}$ | 71.4 | 75.9 | **77.0** |
| Avg | 76.7 | 79.7 | **82.6** |
| RefCOCOg$_{val}$ | 84.7 | 86.1 | **87.3** |
| RefCOCOg$_{test}$ | 83.1 | 84.0 | **86.9** |
| Avg | 83.9 | 85.0 | **87.1** |

**Benchmarks.** We perform a comprehensive evaluation of our model across a diverse suite of benchmarks, spanning counting, fine-grained perception, hallucination, and grounding. Specifically, we assess counting capabilities using CountBench (Paiss et al., 2023) and Pixmo, while fine-grained perception is evaluated via the V* visual search benchmark (Wu & Xie, 2023). To measure hallucination tendencies, we employ MMVP (Tong et al., 2024) and HallusionBench (Guan et al., 2024). Finally, the model's grounding performance is validated on RefCOCO series.

**Baselines.** We compare Visual Para-Thinker against:

- **Qwen2.5-VL:** Direct inference utilizing the original 3B, 7B, and 72B model variants.

- **Sequential:** A standard Chain-of-Thought (CoT) approach to elicit step-by-step reasoning.

- **Majority voting:** A consensus mechanism selecting the most frequent output across multiple independent inferences.

## 4.2. Performance

**Vision-Centric Task** Table 1 and Fig 5(c) show the performance of our method in visual perception tasks. In the counting task, our method achieved an average improvement of 3.7% on the 3B model parameters compared to the original model. On the 7B parameter scale with larger parameters, our method achieved an average improvement of 2.7%. Moreover, Visual Para-Thinker has shown improvements in both the hallucination task such as MMVP and HallusionBench. For the 7B model, our method has improved the performance by nearly 4%. It demonstrates that parallel thinking and reasoning strategies play a significant role in mitigating hallucinations. Furthermore, our method yields substantial enhancements in high-resolution, fine-grained perception tasks, particularly in visual search

*Table 3.* Evaluation results for impact of the number of reasoning paths on Visual Para-Thinker-3B across counting, fine-grained visual perception, and hallucination tasks. **Bold** text indicates the best performance.

| Model | CountBench | Pixmo | V* | MMVP | HallusionBench | Average |
|---|---|---|---|---|---|---|
| Visual Para-Thinker-3B (1 path) | 69.4 | 51.4 | 63.2 | 60.0 | 59.5 | 60.7 |
| Visual Para-Thinker-3B (2 path) | 71.3 | 52.3 | 64.6 | 62.1 | 60.4 | 62.1 |
| Visual Para-Thinker-3B (4 path) | **72.6** | **54.4** | **67.9** | **63.6** | **62.2** | **64.1** |

*Table 4.* Ablation study on the effect of Pa-Attention and LPRoPE. **Bold** text indicates the best performance.

| Pa-Attention | LPRoPE | Counting | | Perception | Grounding | | | Hallucination | |
|---|---|---|---|---|---|---|---|---|---|
| | | CountBench | Pixmo | V* | RefCOCO | RefCOCO+ | RefCOCOg | MMVP | HallusionBench |
| - | - | 69.2 | 50.8 | 63.3 | 84.3 | 76.7 | 83.6 | 59.2 | 56.1 |
| ✗ | ✓ | 70.8 | 52.9 | 63.8 | 85.7 | 79.7 | 85.1 | 61.9 | 58.7 |
| ✓ | ✗ | 70.0 | 51.1 | 64.3 | 87.5 | 80.4 | 84.9 | 58.2 | 57.4 |
| ✓ | ✓ | **72.6** | **54.7** | **67.9** | **88.5** | **82.6** | **87.1** | **63.6** | **62.2** |

*Table 5.* Efficiency comparison on V* benchmark. Majority voting@4 means generating four times.

| Method | Time (s) ↓ | Throughput (token/s) ↑ |
|---|---|---|
| Qwen2.5-VL-3B | 266 | 58 |
| Sequential | 481 | 53 |
| Majority Voting | 1197 | 49 |
| Ours (w/o reuse) | 364 | 104 |
| Ours (w/ reuse) | 312 | 123 |

scenarios, V* benchmark. Notably, our 7B-parameter model delivers a remarkable improvement on V* and hallucination benchmark, surpassing the performance of frontier models, such as GPT-4o, Gemini2.5-Pro, and Qwen2.5-VL-72B despite the significant difference in model size.

**Grounding Task** As demonstrated in Table 2, Visual Para-Thinker-3B achieves an average Acc@0.5 of 88.5% on the RefCOCO benchmark, representing a 4% improvement over Qwen2.5-VL-3B. Furthermore, our method yields gains of approximately 6% and 4% on the RefCOCO+ and Ref-COCOg benchmarks, respectively. Furthermore, the performance of Visual Para-Thinker on the grounding task surpasses that of Sequential, suggesting that parallel reasoning retains significant competitiveness within visual grounding.

### 4.3. Efficiency

In this part, we evaluate the reasoning efficiency of our implementation within the vLLM framework.

**Total time.** Specifically, as demonstrated in Table 5, the inference total time of Visual Para-Thinker is comparable to that of Qwen2.5-VL-3B on the V* benchmark. In contrast, the sequential and majority voting schemes incur significantly higher total inference time. The sequential approach requires 481s, nearly twice that of Qwen2.5-VL-3B, while

the majority voting scheme takes 1197s, representing an almost fourfold increase over the baseline. This demonstrates the superiority of our parallel reasoning method in terms of efficiency. Specifically, reusing the KV cache during the parallel inference stage eliminates the computational overhead typically incurred by re-prefilling during the summarization phase. This optimization ultimately results in a 15% reduction in total inference time.

**Throughput.** Table 5 presents the throughput speed of Visual Para-Thinker-3B. We can observe that our method, which incorporates KV cache reuse, achieves a 2.5x increase in throughput. The inference throughput of Qwen2.5-VL-3B is comparable across Sequential and Maj@4.

### 4.4. Ablation Study

**Pa-Attention** Table 4 illustrates the role of Pa-Attention on Visual Para-thinker. The removal of the Pa-Attention component results in suboptimal performance across all evaluated tasks, including Counting, Perception, Grounding, and Hallucination assessment. This indicates that the absence of Pa-Attention compromises the independence of distinct reasoning paths, leading to mutual interference and representational confusion during training, which ultimately degrades overall performance.

**LPRoPE** Table 4 also demonstrates the significance of LPRoPE. From the table, we can see that LPRoPE has enhanced the reasoning path recognition ability of Visual Para-Thinker. Specifically, on the Pixmo benchmark for the counting task, LPRoPE improved performance by nearly 4%, which indicates that the recognition of different paths can maximize the parallel thinking ability.

**Number of visual reasoning paths** Table 3 illustrates the effects of different reasoning paths. As the number of rea-

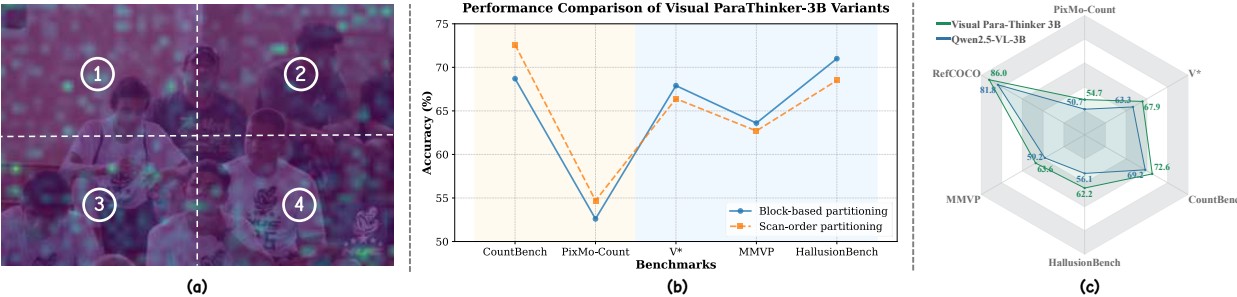

*Figure 5.* (a) depicts the attention allocation patterns observed in the counting task, while (b) compares the performance of the two visual partitioning modes across various visual tasks. (c) demonstrates the superior performance of our method.

soning paths increases, our proposed method demonstrates consistent performance gains in vision-centric tasks. It demonstrates that our method has a certain scaling effect in terms of the number of reasoning paths. Notably, the performance gains on the hallucination and visual search benchmarks are even more pronounced.

## 5. Analysis

In this section, we provide a comprehensive analysis of the advantages and disadvantages associated with the two visual partitioning strategies: Block-based partitioning and Scan-order partitioning. As illustrated in Fig 5(b), the performance of partitioning strategies varies significantly across different task types. In counting-based benchmarks, such as CountBench and PixMo-Count, the Scan-order strategy demonstrates a clear advantage. Conversely, in tasks requiring fine-grained visual reasoning or search—specifically within the V*, MMVP, and HallusionBench datasets—the Block-based strategy consistently outperforms the Scan-order approach.

We attribute this disparity to the nature of attentional allocation during counting tasks, which tends to be highly diffuse, as illustrated in Fig 5(a). Under these conditions, the Scan-order partitioning scheme may inadvertently fragment a single object across non-contiguous processing paths. This fragmentation likely induces perceptual inconsistency, leading to duplicate or erroneous counts and ultimately degrading overall accuracy.

Conversely, by employing the Scan-order partitioning strategy, each processing path maintains access to a comprehensive global receptive field. Rather than fragmenting the visual context, this approach modulates the attentional allocation of the reasoning paths by permuting the sequence of intake. This reorientation of the model's inferential trajectory effectively mitigates the occurrence of object hallucinations. Based on the aforementioned analysis, it is evident that both Block-based and Scan-order partitioning strategies possess distinct strengths and limitations. Consequently, we

have adopted a hybrid training paradigm that integrates both approaches to leverage their complementary advantages to facilitate customized reasoning.

## 6. Related Work

### 6.1. RLVR

The success of DeepSeek-R1 (Guo et al., 2025) has completely driven a significant increase in research attention towards the RLVR paradigm. Subsequent studies have demonstrated RLVR's effectiveness across diverse domains—including math (Guo et al., 2025), coding (Xie et al., 2025), multi-modal reasoning (Li et al., 2025b; Huang et al., 2025; Shen et al., 2025; Huang et al., 2026b; Wang et al., 2025b), and video reasoning (Feng et al., 2025; Li et al., 2026; 2025a; Yang et al., 2025b). Following DAPO (Yu et al., 2025) and GSPO (Zheng et al., 2025a), the algorithms have also continued to improve. However, none of these methods fundamentally changed the architecture of the model's sequential thinking process.

### 6.2. Parallel Generation and Thinking

Although decoder-only LLMs have achieved great success, its extremely slow decoding speed has become a major factor restricting its development. Therefore, a few work begin to explore the generation of multiple tokens at once. Diffusion-based models (Nie et al., 2025; Zhu et al., 2025) can generate multiple tokens in parallel during each diffusion step. However, since it completely abandoned the only-decode architecture, there are still some doubts regarding its applicability and generalization capabilities. In addition, other works, such as Para-Thinker (Wen et al., 2025), maintain the standard language architecture and achieve parallel reasoning solely by generating and caching different paths. However, all these efforts were confined to LLMs. Therefore, we propose the first parallel-thinking MLLM, termed Visual Para-Thinker.

# 7. Conclusion

Our research has proposed the Visual Para-Thinker framework, which is the first parallel reasoning framework in MLLMs that simultaneously generates multiple reasoning paths and integrates multiple thinking paths. For visual parallel reasoning, we proposed Pa-Attention and LPRoPE to distinguish and isolate different paths, and provided an implementation of an inference engine based on vLLM. We have demonstrated the effectiveness of our parallel reasoning in visual tasks through multiple visual perception tasks. Future work can conduct a more in-depth analysis of the differences and connections between parallel reasoning and deep reinforcement learning on visual tasks.

# 8. Rebuttal

## 8.1. Reviewer 1

**Question 1:** Lacks detailed ablation comparing the individual contributions of the two partitioning strategies.

**Answer:** In Section 5 and Fig 5, we analyzed it.

1. Scan-order excels at counting tasks, where sequential visual scanning is beneficial.

2. Block-based is more effective for grounding, hallucination, and visual search tasks, where spatial locality matters.

This complementarity motivates our hybrid training paradigm. Importantly, at inference time, only one strategy is applied per query (selected based on task type), so no redundant computation is incurred.

**Question 2:** Does not fully explore how inference time and memory with an increasing number of reasoning paths.

**Answer: 1. Inference time.** Table 5 compares the inference time of our method (using 4 paths) on the V* benchmark against several baseline approaches. The results indicate that the inference time of our method is comparable to that of the original Qwen2.5-VL. This negligible overhead is primarily attributable to our efficient engineering optimizations.

Furthermore, we evaluated the impact of increasing the inference path length on the inference time based on V*.

| Method | Time (s) ↓ |
|---|---|
| Qwen2.5-VL-3B | 266 |
| Ours (4 paths) | 312 |
| Ours (2 paths) | 294 |
| Ours (1 path) | 281 |

**2. Memory.** We provide some additional analysis on the memory usage of Visual Para-Thinker. During the parallel reasoning stage, the maintenance of KV caches across multiple reasoning paths inevitably increases memory consump-

tion. However, leveraging advanced engineering optimizations—specifically Copy-on-Write, Forking, and KV Cache Reuse—we observed that this overhead remains within 5%, which is well within acceptable limits for practical deployment. Conversely, Test-Time Scaling (TTS) paradigms inherently aim to push the performance boundaries of models by increasing computational and memory overhead.

## 8.2. Reviewer 2

**Question 1:** The improvement diminishes with model scale. For models 100B, it may become negligible as the base performance saturates.

**Answer:** We evaluated our approach using Qwen2.5-VL (3B and 7B) as base models, maintaining consistent training data volumes. When we provided the model of 7B with more training data, its performance further improved. As shown below, our method exhibits robust performance improvements, indicating that its efficacy does not diminish with increased parameter counts.

## 8.3. Reviewer 3

**Question 1:** Is it the distilled data or the components that actually drive the performance in this paper?

**Answer:** In table 4 and 1, our experimental comparison is a comparison of methods under the same data scale. This is sufficient to prove the effectiveness of our method.

**Question 2:** While the empirical results show solid improvements over the base model, sequential CoT, and majority voting, the evaluation lacks comparisons with other advanced inference-time scaling techniques. **Answer:** The latest results have been updated in Table 1.

**Question 3:** The evaluation focuses heavily on dense visual perception tasks, such as counting (Pixmo), grounding (RefCOCO), and hallucination ( HallusionBench). The paper lacks comprehensive evaluation on standard, broad-domain multimodal reasoning benchmarks (e.g., MMMU or V*), leaving the generalizability of this parallel reasoning paradigm across more diverse cognitive tasks in question.

**Answer:** V* results are in Table 1 of the paper. We additionally evaluated on MMMU: Visual Para-Thinker achieves 55.8 (3B, +2.7) and 61.3 (7B, +2.7) over Qwen2.5-VL baselines (53.1 and 58.6). Combined with V* gains (+4.6/+2.9), these results demonstrate generalization beyond dense perception tasks. With data scaling, 7B further reaches 80.4 on V*, suggesting strong potential for broader generalization.

## 8.4. Reviewer 4

**Question 1:** More baseline(SFT and RL).

**Answer:** The latest results have been updated in Table 1.

## Impact Statement

This work advances multimodal machine learning by introducing a parallel reasoning paradigm for MLLMs. It presents a framework to advance multimodal machine learning by introducing a new test-time scaling paradigm. Our work opens significant research potential in combining parallel reasoning with RL scaling, which may lead to emergent capabilities in downstream applications such as embodied intelligence and integrated Diffusion-LLM systems. While these advancements offer broad societal benefits, they also necessitate responsible deployment to manage risks related to privacy and ethical concerns. We believe this research serves as a critical stepping stone toward more robust and efficient multimodal AI.

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

# A. Appendix A.

## A.1. Training Details

This section details the configuration used for training Visual Para-Thinker.

*Table 6.* Training Configuration for Visual Para-Thinker

| Parameter | Value |
|---|---|
| Batch Size | 1 |
| Gradient Accumulation Steps | 8 |
| Learning Rate | $1 \times 10^{-5}$ |
| Training Epochs | 2 |
| Warmup Ratio | 0.1 |
| Learning Rate Scheduler | Constant |
| Max Pixels | 12845056 |
| Hardware | 4 GPUs |

## A.2. Composition of Training Data

We constructed a comprehensive parallel reasoning dataset totaling 163,000 question-answer pairs, specifically curated to enhance the model's capabilities across diverse reasoning dimensions. As illustrated in Fig 6, the dataset is strategically partitioned into three core categories: Grounding, Fine-grained Perception, and Counting.

- **Grounding (67K):** Constituting the largest portion of the dataset, this task is designed to fortify the model's spatial localization capabilities. It integrates data from three cornerstone benchmarks: RefCOCO (21,000 pairs), RefCOCO+ (25,000 pairs), and RefCOCOg (21,000 pairs).

- **Fine-grained Perception (65K):** To foster a deeper understanding of detailed visual attributes and semantic nuances, we incorporated 65,000 samples. This category comprises 35,000 pairs derived from the LAION dataset and 30,000 pairs from COCO, providing a diverse array of high-resolution visual contexts.

- **Counting (31K):** To improve numerical reasoning and object enumeration skills, the dataset includes 31,000 specialized samples sourced from the PixMo-Count dataset.

By synthesizing these diverse tasks and data sources, the resulting training set provides a robust foundation for multi-modal reasoning and precise visual understanding.

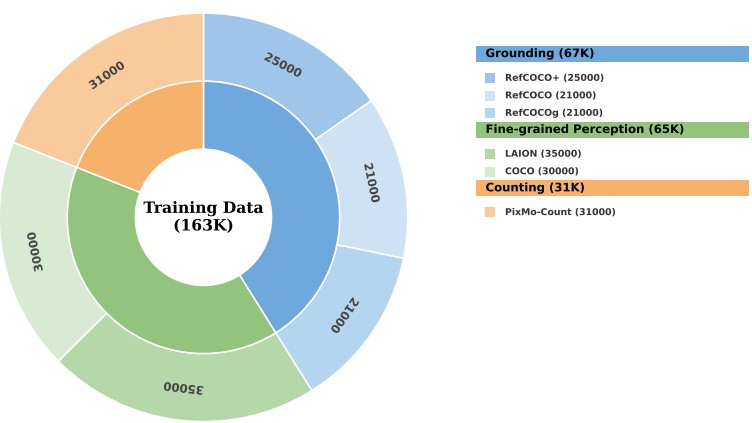

*Figure 6.* Distribution and composition of the 163K parallel reasoning training data across various tasks and benchmarks.

## A.3. Examples of data

The steps for constructing SFT training data after sampling from the teacher model to obtain Block-based partitioning captions and Scan-order partitioning captions are as follows: (1) Connect the same type of partitions to form different reasoning paths; (2) Insert special markers to distinguish parallel reasoning paths; (3) Append the actual final answers. Here are two examples:

---

**Data Example(Block-based partitioning)**

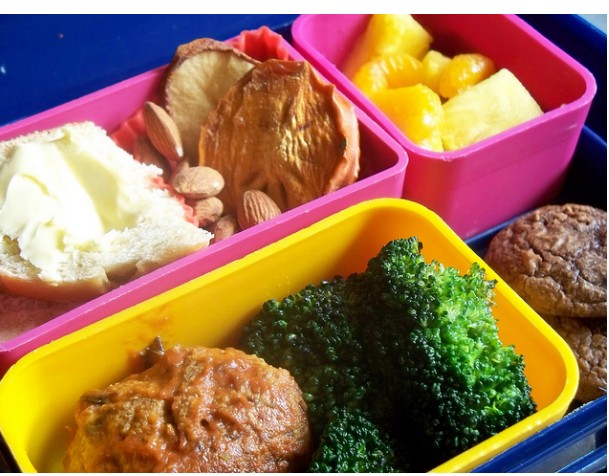

*<|User|><image>What is the color of the container holding the broccoli and meatball? A. yellow B. pink C. blue D. green . This is a multiple-choice question, Please select one option. You FIRST think about the reasoning process as an internal monologue and then summarize the reasoning process to get the final answer. The summary process MUST BE enclosed within <summary> </summary> tags. The final answer MUST BE put in \boxed{}.<|Assistant|><vllm_pad>*

```
<think1>
```
Top-Left: This quadrant shows a pink container holding a slice of white bread with a dollop of white spread, several whole almonds, and two dried fruit slices, one of which is round and brown. The container has a red paper liner.
```
</think1>
```

```
<think2>
```
Top-Right: This quadrant shows a pink container filled with yellow fruit pieces, including what appear to be pineapple chunks and mandarin orange segments. The container is set against a blue background.
```
</think2>
```

```
<think3>
```
Bottom-Left: This quadrant shows a yellow container holding a round, brown meatball covered in red sauce and a large floret of green broccoli. The container has a yellow rim.
```
</think3>
```

```
<think4>
```
Bottom-Right: This quadrant shows a dark brown, round cookie with a textured surface. Part of a blue container edge is visible, and another similar cookie is partially visible behind the first one.

```
<summary>
```
By analyzing multiple reasoning processes above, I concluded that: The final answer is $\boxed{\text{A. yellow}}$.
```
</summary>
```

---

**Data Example(Scan-order partitioning)**

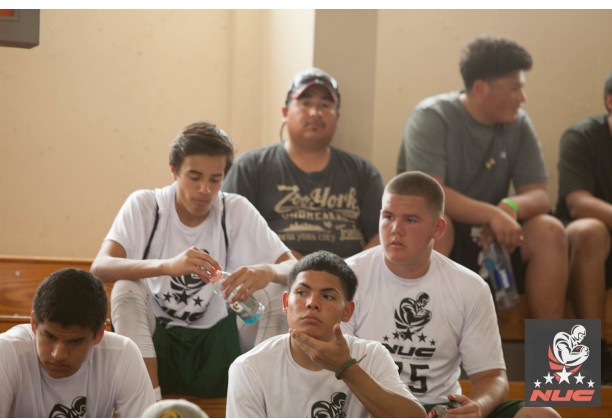

*<|User|><image>How many people are in the image? You FIRST think about the reasoning process as an internal monologue. In your thoughts, try to locate and count the objects one by one or group by group to ensure accuracy. Then, summarize the counting process to get the final number. The summary process MUST BE enclosed within <summary> </summary> tags. The final answer MUST BE a single integer put in \boxed{}.<|Assistant|><vllm_pad>*

```
<think1>
```
Scanning Left-to-Right: 1. Found a young man in a white shirt looking down on the far left. That makes 1. 2. Next, a young man in a white shirt pouring water from a bottle. That makes 2. 3. Behind him, a man in a dark grey 'Zoo York' t-shirt and a cap. That makes 3. 4. In the center foreground, a young man in a white shirt with his hand on his chin. That makes 4. 5. To his right, a young man in a white shirt with a 'NUC' logo. That makes 5. 6. Behind him, a young man in a grey t-shirt looking to the right. That makes 6. 7. On the far right, a partially visible person in a dark shirt. That makes 7. Total count: 7.
```
</think1>
```

```
<think2>
```
Scanning Top-to-Bottom: Starting from the top, I first encounter the man in the 'Zoo York' t-shirt and cap. That makes 1. To his right, the young man in the grey t-shirt. That makes 2. To the far right, the partially visible person in a dark shirt. That makes 3 ......
```
</think2>
```

```
<think3>
```
Scanning Right-to-Left: Starting from the far right, I see a partially visible person in a dark shirt. That makes 1. Moving left, I see a young man in a grey t-shirt. That makes 2. Next, a young man in a white 'NUC' shirt. That makes 3. In front of him, a young man in a white shirt with his hand on his chin. That makes 4. Behind him, a man in a 'Zoo York' t-shirt and cap. That makes 5 ......
```
</think3>
```

```
<think4>
```
Scanning Bottom-to-Top: Starting from the bottom, I first find the young man looking down on the far left. That makes 1. Moving up, I see the young man with his hand on his chin. That makes 2. To his right, the young man in the 'NUC' shirt. That makes 3. Above him, the young man in the grey t-shirt ......
```
</think4>
```

```
<summary>
```
By integrating the detailed scans with a global view of the image to avoid double-counting or omissions, I have verified the count. The final answer is 7 .
```
</summary>
```

# B. Appendix B.

## B.1. LPRoPE

Standard Rotary Positional Embedding (RoPE) exhibits limitations in multi-path scenarios, such as hierarchical summarization. As illustrated in Fig 7(a), tokens from distinct paths ($T_{1,t}$ and $T_{2,t}$) sharing the same relative index $t$ receive identical rotation matrices. This results in positional collisions, rendering the model unable to distinguish token provenance during the aggregation stage.

To address this, we propose LPRoPE, which injects learnable path embeddings ($e_1, e_2$) into the latent representations prior to the rotation operation. Geometrically, this pre-rotation shift ensures that tokens sharing the same rotation angle $\theta$ remain in distinct, quasi-orthogonal subspaces, as shown in Fig7(b). By adaptively optimizing these path-specific offsets, LPRoPE guarantees path discriminability and robust identifiability throughout the attention mechanism.

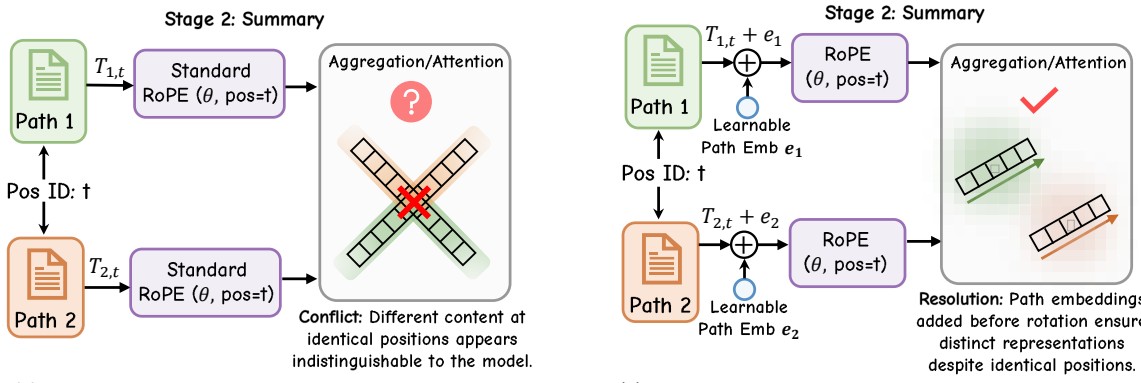

*Figure 7.* Comparison between Standard RoPE and LPRoPE. (a) Standard RoPE applies identical rotations to tokens at the same index across different paths, causing representation ambiguity. (b) LPRoPE adds learnable path embeddings ($e_i$) before rotation, shifting features into distinct geometric regions to preserve path discriminability.

## B.2. Visual Para-Thinker workflow

In this section, we describes the workflow of Visual Para-Thinker as Algorithm 1.

---

**Algorithm 1** Parallel Reasoning and Aggregation Framework

---

**Input:** Image $I \in \mathbb{R}^{H \times W \times C}$, Instruction $Q = \{q_1, \ldots, q_L\}$
**Output:** Final Answer $\mathcal{A}$

---

**STAGE 1: Parallel Reasoning**
$C_{shared} \leftarrow \text{Encoder}(I, Q)$          {Construct Shared Multimodal Context}
$\{I_1, I_2, \ldots, I_N\} \leftarrow \text{Partition}(I, N)$          {Visual Partitioning, $N = 4$}
**for** $k = 1$ **to** $N$ **in parallel do**
     $\text{Attn}_k \leftarrow \text{Pa-Attention}(C_{shared}, I_k)$          {Branch $k$ Isolation}
     $P_k \leftarrow \text{MLLM}(\text{Attn}_k, \texttt{<think } k\texttt{>}) + \text{LPRoPE}$          {Generate Reasoning Path $k$}
**end for**

---

**STAGE 2: Summary**
$\mathcal{P} \leftarrow \text{Merge}(\{P_1, P_2, \ldots, P_N\})$          {Collect all reasoning paths}
$S_{agg} \leftarrow \text{LPRoPE}(\mathcal{P} \mid C_{shared})$          {Synthesize with LPRoPE & Aggregation}
$\mathcal{A} \leftarrow \text{Decode}(S_{agg})$          {Final Answer Generation}
**return** $\mathcal{A}$

---

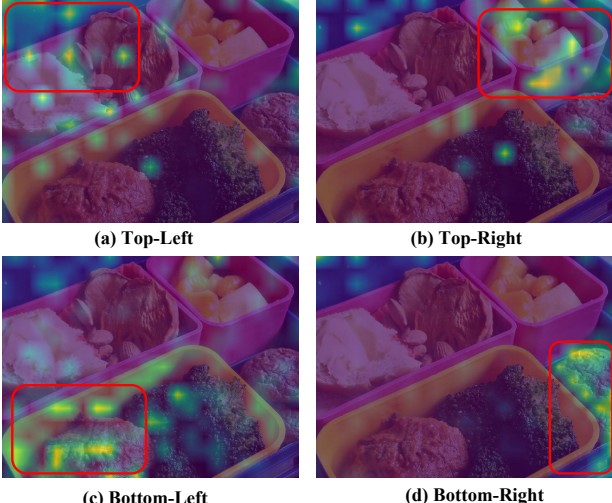

*Figure 8.* Visualization of attention maps across parallel reasoning pathways. Heatmaps and corresponding bounding boxes highlight how different paths focus on diverse, complementary image regions to facilitate joint visual reasoning.

### B.3. Attention Distribution in Visual Reasoning

To understand the specialized focus of the Visual Para-Thinker, we visualize the attention weights of the final reasoning tokens over visual regions. We average the attention maps from the last layer across all parallel reasoning paths.

As illustrated in Fig 8, different reasoning pathways focus on distinct, task-relevant image regions. This diversity suggests that parallel paths capture complementary visual information, effectively decomposing the visual reasoning task to support a more robust final decision.

### B.4. Layer-wise Attention Dynamics

To investigate the internal reasoning of Visual Para-Thinker-7B, we visualize attention maps across all 28 layers under two visual partitioning strategies. As shown in Fig 9, the model adapts its attention mechanism to the input structure: Block-based partitioning Fig 9 (a) promotes localized feature extraction, while Scan-order partitioning Fig 9 (b) facilitates global contextual integration.

### B.5. Details of data construction

Based on an analysis of the two distinct visual partitioning strategies, we implemented a hybrid training scheme that integrates both approaches. Specifically, we applied Scan-order strategy to data involving counting tasks, while Block-based strategy was utilized for all other tasks. Drawing an analogy to classical computational paradigms, we conceptualize Block-based strategy as a branching recursive strategy and Scan-order strategy as a search-scanning strategy. By training on this mixed framework, we achieved state-of-the-art performance and established a versatile foundation for multi-strategy inference.

Furthermore, to mitigate hallucination during data construction within Block-based strategy, we utilized a teacher model to generate the training data. We employed four discrete sub-images; since these sub-images were spatially isolated, the reasoning paths generated by the teacher model remained independent of one another. We then utilized these decoupled reasoning trajectories to train our model, Visual Para-Thinker. This data construction methodology is analogous to the 'Teacher-Forcing' paradigm commonly used in sequence training.

Through this approach, we ensure alignment between the data generation and the training phase. Furthermore, by enhancing the vLLM framework, we have bridged the gap between training and inference. Consequently, our methodology achieves end-to-end consistency across data construction, model training, and inference.

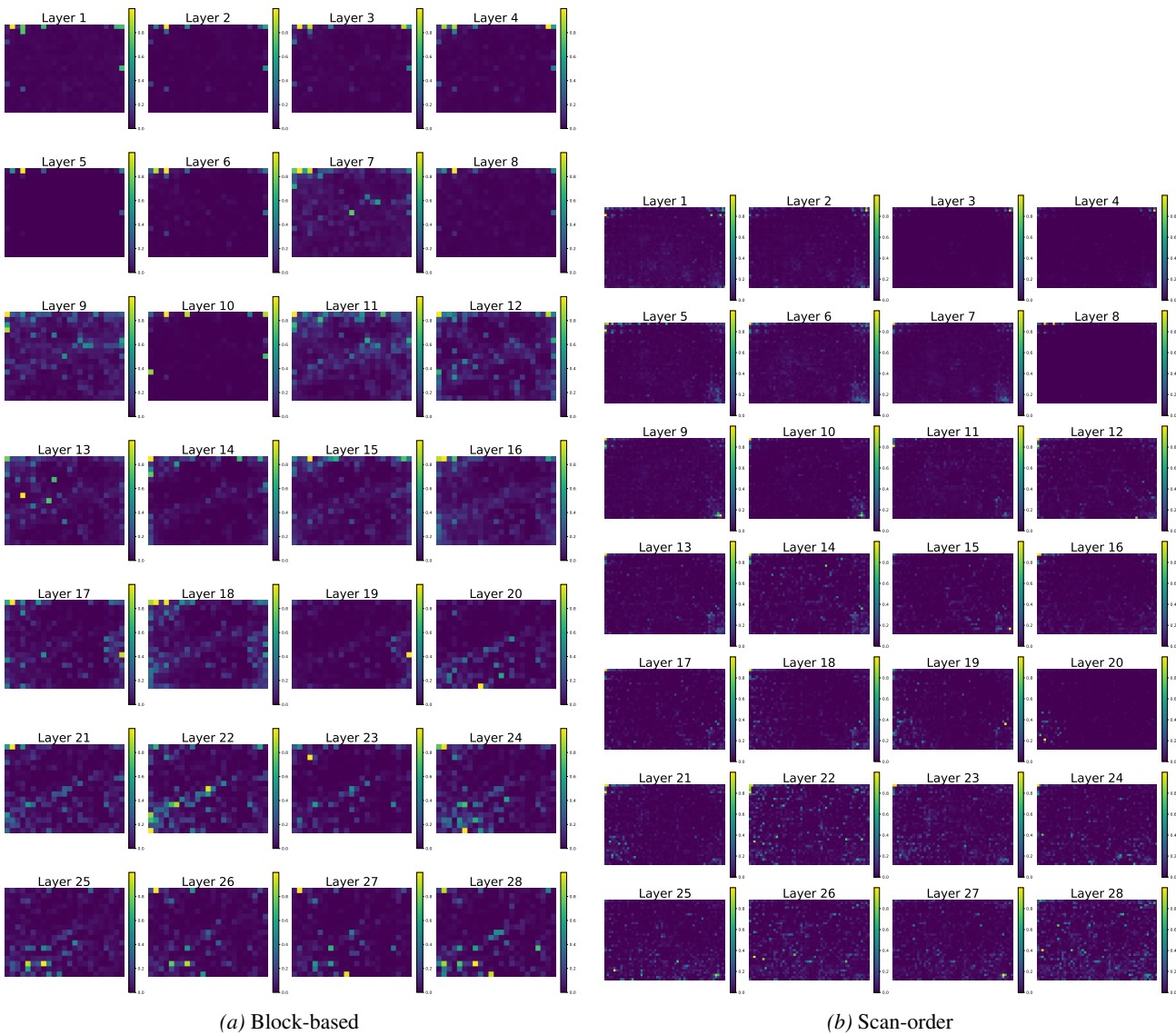

*(a)* Block-based        *(b)* Scan-order

*Figure 9.* Evolution of layer-wise attention (Layers 1–28) for different visual partitioning strategies. (a) Block-based: attention remains concentrated on contiguous image regions, favoring local reasoning. (b) Scan-order: attention maps exhibit a more diffuse, globalized distribution, supporting broad feature integration. Intensity values are averaged across all visual tokens.

