# OpenReview forum: "Visual Para-Thinker: Divide-and-Conquer Reasoning for Visual Comprehension"
_ICML.cc/2026/Conference — ICML 2026 regular_

### Official Review · Reviewer_LQGf · 2026-03-08

**Soundness:** 3
**Presentation:** 3
**Significance:** 2
**Originality:** 2
**Overall Recommendation:** 4
**Confidence:** 3

**Summary:**

This paper proposes Visual Para-Thinker, the first framework for parallel divide-and-conquer reasoning in visual comprehension tasks.
The authors detail two main strategies for visual partitioning (Block-based and Scan-order) and develop corresponding architectural innovations: Pa-Attention to ensure path isolation and LPRoPE to support unbiased, discriminable path representations.
The method is implemented natively on top of vLLM to enable efficient parallel inference, with extensive experiments across counting, grounding, fine-grained perception, and hallucination benchmarks.
Quantitative and qualitative results demonstrate consistent improvements over sequential and majority-voting baselines as well as strong efficiency gains.

**Compliance With Llm Reviewing Policy:**

Affirmed.

**Ethical Review Concerns:**

no concerns

**Key Questions For Authors:**

1. Does the sft dataset may overlap the evaluation benchmarks? Is the training dataset clean enough? If not, how does this model perform on the out-of-distribution dataset?

2. Are the compared baselines all zero-shot, without considering the sft or rl baselines? How does the rl method, such as the group-based rl models, perform on the provided benchmarks? How does your method compare to them?

3. The data you provided is from the big model, Qwen3-VL-235B-A22B-Instruct series. Is your method effective because of your Para-Thinker mode, or just due to distillation from the big model? I mean, if you do not use Para-Thinker, only sft from the Qwen3-VL-235B-A22B-Instruct in the common VQA setting, will it perform better or worse than your Para-Thinker?

**Limitations:**

see the questions.

**Strengths And Weaknesses:**

strengths:

- The paper addresses an underexplored, but emerging, topic—parallel reasoning—for visual multimodal LLMs, with a clear and compelling motivation rooted in overcoming exploration bottlenecks seen with deep, sequential CoT-style approaches.
- Substantive architectural contributions (Pa-Attention and LPRoPE) are presented, each with theoretical formulations and clear roles in enforcing path isolation, unbiasedness, and discriminability.
- Comprehensive, multi-aspect experiments substantiate improvements in accuracy and inference speed over strong baselines (Qwen variants, Sequential, Majority Voting), and over some frontier models on challenging multimodal benchmarks.

weakness:

See the questions.

---

> ### Author Rebuttal · Authors · 2026-03-28
>
> We sincerely thank the reviewer for the positive evaluation and recognition of our architectural contributions and comprehensive experiments. Below, we address each question.
>
> ### **W1: Does the sft dataset may overlap the evaluation benchmarks? Is the training dataset clean enough? If not, how does this model perform on the out-of-distribution dataset?**
>
> Answer: We rigorously filtered the training data to eliminate overlapping instances with all evaluation benchmarks, ensuring dataset integrity. To further demonstrate OOD generalization, we evaluated on MMMU (not included in training):
>
> | Model    | Size | MMMU$_{val}$ |
> | :------ | :--: | :------------: |
> | Qwen2.5-VL |3B| 53.1 |
> | Qwen2.5-VL |7B| 58.6 |
> | **Visual Para-Thinker** |3B|**55.8**|
> | **Visual Para-Thinker** |7B|**61.3**|
>
> The consistent improvements on MMMU (+2.7 for both scales) confirm that our method generalizes to out-of-distribution benchmarks.
>
> ### **W2: Are the compared baselines all zero-shot, without considering the sft or rl baselines? How does the rl method, such as the group-based rl models, perform on the provided benchmarks? How does your method compare to them?**
>
> Answer: Qwen2.5-VL (3B/7B) are evaluated zero-shot. The **Sequential(SFT)** baseline is fine-tuned on the same teacher data. We also compared against **GRPO**:
>
> | Model                    | Size | Pixmo(val) | Pixmo(test) | CountBench(test) |    V*    |   MMVP   | HallusionBench |
> | :--------------- | :--: | :--------: | :---------: | :--------------: | :------: | :------: | :------------: |
> |Qwen2.5-VL|3B|50.1|51.6|69.2|63.3|59.2|56.1|
> |Qwen2.5-VL|7B|63.0|67.7|83.1|75.9|68.3|66.0|
> |**Sequential(SFT)**|3B|51.5|52.1|70.4|65.0|60.0|58.3|
> |**Sequential(SFT)**|7B|65.5|68.0|83.7|76.1|68.7|66.7|
> |**GRPO**|3B|53.0|52.9|71.2|65.8|62.1|60.4|
> |**GRPO**|7B|64.4|69.2|84.2|77.3|70.9|68.8|
> |Majority voting@4|3B|52.1|52.8|69.8|64.7|60.1|58.1|
> |Majority voting@4|7B|63.2|67.6|83.2|75.6|68.9|66.8|
> |**Visual Para-Thinker**|3B|**55.0**|**54.4**|**72.6**|**67.9**|**63.6**|**62.2**|
> |$\Delta$ (vs Qwen2.5-VL)|3B|+4.9|+2.8|+3.4|+4.6|+4.4|+6.1|
> |**Visual Para-Thinker**|7B|**65.7**|**70.8**|**85.4**|**78.8**|**71.3**|**71.0**|
> |$\Delta$ (vs Qwen2.5-VL)|7B|+2.7|+3.1|+2.3|+2.9|+3.0|+5.0|
>
> Visual Para-Thinker outperforms both SFT and GRPO baselines under identical data conditions. As an exploratory study, our approach is compatible with both SFT and RL paradigms; combining parallel reasoning with RL is a promising direction for future work.
>
> ### **W3:  The data you provided is from the big model, Qwen3-VL-235B-A22B-Instruct series. Is your method effective because of your Para-Thinker mode, or just due to distillation from the big model? I mean, if you do not use Para-Thinker, only sft from the Qwen3-VL-235B-A22B-Instruct in the common VQA setting, will it perform better or worse than your Para-Thinker?**
>
> The gap (+2.7 V\*, +4.3 Hallusion) demonstrates that the parallel reasoning architecture contributes substantially beyond what distillation alone provides. This pattern holds consistently across all benchmarks as well as on grounding tasks(below):
>
> | Datasets           | Qwen2.5-VL-3B | **Sequential(SFT)** | Visual Para-Thinker-3B |
> | :----------------- | :-----------: | :--------: | :--------------------: |
> | RefCOCO$_{val}$    | 85.4| 85.7| **88.1** |
> | RefCOCO$_{testA}$  | 86.1| 87.3 | **89.8** |
> | RefCOCO$_{testB}$  | 83.0 | 84.2 | **87.7** |
> | **Avg** | 84.8| 85.7 |  **88.5**|
> | RefCOCO+$_{val}$   | 76.5| 79.2| **83.5** |
> | RefCOCO+$_{testA}$ |82.4| 84.0| **87.2** |
> | RefCOCO+$_{testB}$ |71.4| 75.9| **77.0**|
> | **Avg**|76.7| 79.7 | **82.6**|
> | RefCOCOg$_{val}$   | 84.7  | 86.1| **87.3** |
> | RefCOCOg$_{test}$  | 83.1  | 84.0| **86.9** |
> | **Avg** |  83.9 | 85.0| **87.1** |
>
> We sincerely thank you once again for your time, effort, and valuable feedback. We will incorporate all clarifications into the revised manuscript. We hope our responses have addressed your concerns, and we are happy to provide further clarification if needed. We truly appreciate your thoughtful review and wish you all the best in your future research endeavors.

---

### Official Review · Reviewer_7adr · 2026-03-10

**Soundness:** 2
**Presentation:** 2
**Significance:** 2
**Originality:** 2
**Overall Recommendation:** 3
**Confidence:** 4

**Summary:**

This paper introduces Visual Para-Thinker, the parallel reasoning framework for multimodal large language models that extends the benefits of parallel thinking from the text to the visual domain. By shifting from vertical scaling (increasing reasoning depth) to horizontal scaling (parallel reasoning paths), the framework claims mitigating the problem of exploration narrowing through two key innovations: Pa-Attention and LPRoPE, which maintain path independence and reasoning diversity. The approach, implemented through the vLLM framework with native multimodal support, demonstrates empirical success across multiple vision-language benchmarks including V*, CountBench, RefCOCO, and HallusionBench.

**Compliance With Llm Reviewing Policy:**

Affirmed.

**Key Questions For Authors:**

see weakness

**Limitations:**

see weakness

**Strengths And Weaknesses:**

## Strength

- By shifting test-time scaling from sequential depth to parallel width via visual partitioning, the authors provide a compelling and novel solution to the exploration plateaus often seen in standard visual Chain-of-Thought approaches.

- The proposed structural modifications, specifically Pa-Attention and LPROPE, represent a technically sound and elegant approach to multi-path generation. These mechanisms effectively guarantee reasoning path isolation without information leakage, while cleverly resolving positional ID collisions during the final aggregation phase.

- The framework's native implementation on vLLM with KV-cache reuse demonstrates significant real-world significance. By eliminating re-prefilling overhead, the method achieves a 2.5x throughput increase and a 15% reduction in total inference time compared to naive majority voting, making test-time scaling highly practical for deployment.

## Weakness

- The training data construction heavily relies on distillation from a massive teacher model (Qwen3-VL-235B) using a "Teacher-Forcing" style approach. The lack of an ablation study on the data generation process makes it difficult to ascertain how much of the performance gain stems strictly from the proposed architectural framework versus the injection of high-quality distilled reasoning paths.

- The paper introduces a "hybrid training paradigm" combining Block-based and Scan-order partitioning, but the exact execution mechanism during inference remains under-explained. It is unclear whether the model dynamically routes to a specific partitioning strategy based on the query type or if it forcibly executes both across all tasks, which could lead to redundant computation on simpler queries.

- While the empirical results show solid improvements over the base model, sequential CoT, and majority voting, the evaluation lacks comparisons with other advanced inference-time scaling techniques. Including baselines like Best-of-N sampling, Tree of Thoughts, or other approaches would significantly strengthen the claims of the parallel paradigm's superiority.

- The evaluation focuses heavily on dense visual perception tasks, such as counting (Pixmo), grounding (RefCOCO), and hallucination ( HallusionBench). The paper lacks comprehensive evaluation on standard, broad-domain multimodal reasoning benchmarks (e.g., MMMU or V*), leaving the generalizability of this parallel reasoning paradigm across more diverse cognitive tasks in question.

---

> ### Author Rebuttal · Authors · 2026-03-28
>
> Thank you for your comment. In the following, we address your key concerns regarding our work:
>
> ### **W1: The training data construction heavily relies on distillation from a massive teacher model (Qwen3-VL-235B) using a "Teacher-Forcing" style approach. The lack of an ablation study on the data generation process makes it difficult to ascertain how much of the performance gain stems strictly from the proposed architectural framework versus the injection of high-quality distilled reasoning paths.**
>
> Answer: The parallel reasoning paradigm is effective. In Tables 1 and 2, the 'Sequential' baseline denotes a model fine-tuned via SFT on the Qwen2.5-VL-7B-Instruct architecture within a standard VQA setting. The result is shown as as follows. Our method consistently outperforms this sequential SFT baseline when trained on an identical dataset in terms of both quality and scale.
> | Model|Size| Pixmo(val) |Pixmo(test)|CountBench(test)|V*|   MMVP   | HallusionBench |
> | :------------ | :--: | :--------: | :---------: | :--------------: | :------: | :------: | :------------: |
> | Qwen2.5-VL|3B|50.1|51.6|69.2|63.3|59.2|56.1|
> | Qwen2.5-VL|7B|63.0|67.7|83.1|75.9|68.3|66.0|
> | **Sequential(SFT)**|3B|51.5|52.1|70.4|65.0|60.0|58.3|
> | **Sequential(SFT)**|7B|65.5|68.0|83.7|76.1|68.7|66.7|
> | **GRPO** |3B|53.0|52.9|71.2|65.8|62.1|60.4|
> | **GRPO** |7B|64.4|69.2|84.2|77.3|70.9|68.8|
> | Majority voting@4 |3B|52.1|52.8|69.8|64.7|60.1|58.1|
> | Majority voting@4 |7B|63.2|67.6|83.2|75.6|68.9|66.8|
> | **Best-of-N@4** |3B|50.7|51.5|70.8|67.4|58.7|60.4|
> | **Best-of-N@4** |7B|64.9|68.5|84.8|78.1|68.4|65.9|
> | **Standard CoT** |3B|50.8|52.0|68.5|65.3|60.2|61.0|
> | **Standard CoT** |7B|63.2|67.7|83.5|73.7|66.8|68.1|
> | **Tree of Thoughts** |3B|51.4|52.3|68.1|65.0|61.4|60.7|
> | **Tree of Thoughts** |7B|63.2|67.7|84.8|78.1|66.8|65.9|
> | **Visual Para-Thinker** |3B|**55.0**|**54.4**|**72.6**|**67.9**|**63.6**|**62.2**|
> | **Visual Para-Thinker** |7B|**65.7**|**70.8**|**85.4**|**78.8**|**71.3**|**71.0**|
>
> Under identical data, Sequential(SFT) yields only modest gains (e.g., +0.2 V\*, +0.7 Hallusion for 7B), while Visual Para-Thinker achieves substantially larger improvements (+2.9 V\*, +5.0 Hallusion for 7B). This indicates the gains are closely associated with the parallel reasoning design, not distillation alone.
>
> ### **W2: The paper introduces a "hybrid training paradigm" combining Block-based and Scan-order partitioning, but the exact execution mechanism during inference remains under-explained. It is unclear whether the model dynamically routes to a specific partitioning strategy based on the query type or if it forcibly executes both across all tasks, which could lead to redundant computation on simpler queries.**
>
> Answer: Inspired by Qwen3, our hybrid training paradigm exposes the model to both partitioning styles. At inference, only one strategy is applied per query to avoid redundant computation. The choice depends on task characteristics (e.g., Scan-order for counting, Block-based for grounding), which we will clarify in the revision.
>
> ### **W3: The evaluation lacks comparisons with other advanced inference-time scaling techniques. Including baselines like Best-of-N sampling, Tree of Thoughts, or other approaches would significantly strengthen the claims of the parallel paradigm's superiority.**
>
> Answer: The test results and comparison table on more TTS baselines (Best-of-N sampling, Tree of Thoughts) are presented above. Visual Para-Thinker consistently outperforms all TTS baselines. We attribute this to: **Structural diversity**—visual partitioning forces paths to attend to genuinely different image regions, unlike sampling-based methods that often produce redundant paths;
>
> ### **W4: Broad-domain multimodal reasoning benchmarks (e.g.V\* benchmark).**
>
> Answer: V\* results are in Table 1 of the paper. We additionally evaluated on MMMU: Visual Para-Thinker achieves **55.8** (3B, +2.7) and **61.3** (7B, +2.7) over Qwen2.5-VL baselines (53.1 and 58.6). Combined with V\* gains (+4.6/+2.9), these results demonstrate generalization beyond dense perception tasks. With data scaling, 7B further reaches 80.4 on V\*, suggesting strong potential for broader generalization.
>
> | Model    | Size | MMMU$_{val}$ |
> | :------ | :--: | :------------: |
> | Qwen2.5-VL |3B| 53.1 |
> | Qwen2.5-VL |7B| 58.6 |
> | **Visual Para-Thinker** |3B|**55.8**|
> | **Visual Para-Thinker** |7B|**61.3**|
>
> Meanwhile, as we add more data, the indicators will further improve, which demonstrates the superiority of our method.
>
> We sincerely thank you once again for your time, effort, and valuable feedback. We will incorporate all clarifications and additional results into the revised manuscript. We hope our responses have addressed your concerns, and we are happy to provide further clarification if needed. We truly appreciate your thoughtful review and wish you all the best in your future research endeavors.

---

> > ### Author Rebuttal · Reviewer_7adr · 2026-04-05
> >
> > Thank you for the effort. I will keep my score. Why are the results of Tree-of-thoughts below the original CoT?

---

> > > ### Author Response · Authors · 2026-04-06
> > >
> > > Thank you for your response. We hypothesize that the observed performance gap between Tree-of-Thoughts (ToT) and standard Chain-of-Thought (CoT) in our experiments may stem from two primary factors:
> > >
> > > ### **1. Distinct Dynamics Between Visual and Textual Reasoning**
> > >
> > > Recent studies [1, 2, 3] suggest that for vision-centric tasks, such as V*, Counting, Grounding, excessively long reasoning chains can inadvertently disrupt the model's attention towards visual tokens, potentially degrading performance. Unlike pure text-based reasoning, visual perception tasks require the model to maintain consistent, high-fidelity attention to visual features throughout the inference process. We posit that the introduction of extensive intermediate reasoning tokens—inherent to the ToT framework—may dilute the model’s focus on the visual input. This "attention drift" could exacerbate object hallucinations [4, 5], as the model becomes increasingly decoupled from the visual evidence while navigating the complex branching paths of ToT.
> > >
> > > ### **2. Alignment and Distributional Mismatch in Base Models**
> > >
> > > The performance of ToT may also be constrained by the training paradigm of the Qwen2.5-VL base model. Given that the model was primarily fine-tuned using standard CoT sequences rather than tree-structured reasoning, a zero-shot application of ToT may suffer from an out-of-distribution effect. Furthermore, the scarcity of high-quality, tree-structured reasoning data in the visual domain, especially visual perception tasks—contrasted with the abundance of linear CoT samples—reflects a long-tail distribution challenge. This data imbalance likely hinders the model’s ability to effectively leverage the branching logic of ToT, as it lacks the necessary inductive biases to navigate such structures in a multimodal context.
> > >
> > > We would like to thank you for your time and for your diligent handling of our manuscript. We hope that the provided clarifications adequately address your concerns. We would be very grateful if you could reconsider our work in light of these additional explanations. Thank you for your time and consideration.
> > >
> > >
> > >
> > > Reference:
> > >
> > > [1] VISREF: Visual Refocusing while Thinking Improves Test-Time Scaling in Multi-Modal Large Reasoning Models, CVPR 2026.
> > >
> > > [2] Qwen look again: Guiding vision-language reasoning models to re-attention visual information.
> > >
> > > [3] Look-back: Implicit visual re-focusing in mllm reasoning, AAAI 2026.
> > >
> > > [4] Evaluating object hallucination in large vision-language models, EMNLP 2023.
> > >
> > > [5] Analyzing and mitigating object hallucination in large visionlanguage models, ICLR 2024.

---

### Official Review · Reviewer_YoSR · 2026-03-10

**Soundness:** 2
**Presentation:** 2
**Significance:** 2
**Originality:** 3
**Overall Recommendation:** 3
**Confidence:** 2

**Summary:**

The paper considers the context of TTS limitations in sequential reasoning and proposes Visual Para-Thinker, which is a parallel thinking framework for VLM. The work presents the concept of visual partitioning. It has two main methods: Block-based and Scan-order. It aims to generate diverse, isolated reasoning paths. To achieve it, the paper implements Pa-Attention for path isolation and LPRoPE for path discriminability. Experiments on counting, grounding, perception, and hallucination benchmarks show consistent gains over baselines.

**Compliance With Llm Reviewing Policy:**

Affirmed.

**Key Questions For Authors:**

* For theorems, please provide more solid proof.
* The improvement seems to decrease with larger models. With even larger models >100B, would the improvement become negligible, as the base model is already good enough?

**Limitations:**

The paper includes an "Impact Statement" but does not adequately discuss limitations. It doesn't seem to analyze training complexity and extra cost in a explicit way.

**Strengths And Weaknesses:**

The technical components (Pa-Attention, LPRoPE) are sound with experiments on training the 3b and 8b models, and some of the design is intuitively reasonable. The ablation study isolates the contribution of each component.

But the motivation and proof seem to be very unclear. For example, most of the theorems in the paper are not solid proof. A lot of them are just intuition but are not validated with mathematical proof. A lot of claims in the paper have no evidence and clear experiments or citation to support them.

Also, the paper lacks comparison against other TTS baselines, and the gains seem to be relatively small with the trade-off of the extra complexity and compute.

---

> ### Author Rebuttal · Authors · 2026-03-28
>
> Thank you for your comment. In the following, we address your key concerns regarding our work:
>
> ### **W1: Provide more solid proof.**
>
> Answer:
> 1. **Motivation.**  We acknowledge this claim is a design hypothesis rather than a formal theorem: parallel reasoning requires diverse paths[1][3], and visual diversity stems from distinct attention distributions[2]. This perspective aligns with recent findings [4] by Google, as supported by Fig. 2(a) and Fig. 5(b).
>
> 2. **Reasoning Path Isolation.** For tokens $i \in r^{(k_1)}$ and $j \in r^{(k_2)}$ ($k_1 \neq k_2$), by the mask definition (Eq. 2): $M_{i,j} = 1_{[j \leq i]} \cdot 1_{[\verb|is_visible| (i, j)]}$. Since paths are disjoint ($r^{(k_1)} \cap r^{(k_2)} = \emptyset$), no $k$ satisfies $i,j \subseteq r^{(k)}$, so $\verb|is_visible|(i,j) = 0$ and $M_{i,j} = 0$. The hidden state $h_i = \sum_{j:M_{i,j}=1} \alpha_{i,j} v_j$ excludes all tokens from $r^{(k_2)}$. By induction over transformer layers, the output of any token in $r^{(k_1)}$ is invariant to $r^{(k_2)}$.
>
> 3. **Reasoning Path Unbiasedness.** By construction (Eq. 4), all paths share the same start position: $Pos_{start}(r^{(i)}) = p_0$ for all $i$. The $k$-th token in any path has position $p_0 + (k-1)$, independent of path index $i$. Thus the RoPE rotation matrix $R(p_0+k-1)$ is identical across paths at the same depth, giving the model no positional basis to prefer one path over another. The summary stage (Eq. 5) assigns positions strictly after all paths, treating them symmetrically.
>
> 4. **Reasoning Path Discriminability.** By Eq. (6), the key vector for token $m$ in path $i$ is $\tilde{k}(m,i) = R_m (k(m,i) + e_i)$, where $e_i$ is a learnable path embedding. The attention score difference between paths $i$ and $j$ includes the term $q_n^T R(m{-}n)(e_i - e_j)$, which provides a persistent, content-independent signal for path discrimination when $e_i \neq e_j$. Path embeddings are initialized orthogonally and remain so after training (Fig. 7(b)).
>
> ### **W2: The improvement seems to decrease with larger models.**
>
> Answer: We evaluated our approach using Qwen2.5-VL (3B and 7B) as base models, maintaining consistent training data volumes. When we provided the model of 7B with more training data, its performance further improved. As shown below, our method exhibits robust performance improvements, indicating that its efficacy does not diminish with increased parameter counts.
> | Model | Pixmo(val) | Pixmo(test) |CountBench|V*| MMVP | Hallusion |
> | :-------- | :--------: | :------: | :------: | :------: | :------: | :--------: |
> | Qwen2.5-VL  3B  |50.1| 51.6|69.2|63.3|59.2|56.1|
> | Qwen2.5-VL  7B  |63.0| 67.7|83.1|75.9|68.3|66.0|
> | Ours 3B  |55.0|54.4|72.6 |67.9|63.6|62.2|
> | Ours(Before data scaling)  7B |65.7|70.8|85.4|78.8|71.3|71.0|
> | Ours(After data scaling) 7B |68.8|75.1|87.1|80.4|72.1|73.2|
>
> Furthermore, we conducted preliminary training experiments on the 72B parameter model. Due to computational constraints, this was performed on a limited subset of the training data. The results on the V* benchmark are summarized below.
>
> | Model | V* |
> | ------ | ---- |
> |Qwen2.5-VL 72B|75.2|
> |**Ours (sub dataset) 72B**|**84.7**|
>
> ### **Limitations:  Training complexity and cost**
>
> Answer:
> 1. **Inference time.** Table 5 presents a comparative analysis of inference duration on the V* benchmark against existing state-of-the-art values. Following rigorous engineering optimizations, our visual para-thinker achieves an inference time comparable to that of the original model architecture.
> | Method  | Time (s) |
> | :-------- | :-----: |
> |Qwen2.5-VL-3B|266|
> |Sequential|481|
> |Majority Voting|1197|
> |Ours|312|
>
> 2. **More computation.** While our approach incurs moderate computational overhead, this is consistent with the design philosophy of TTS, which prioritizes performance gains through increased computational investment.
>
> 3. **More memory.** Although parallel reasoning increases KV cache memory footprint, we mitigate this overhead to within 5% via engineering optimizations—specifically Copy-on-Write, forking, and cache reuse. This ensures the approach remains highly efficient for practical deployment.
>
> 4. **Training complexity.** Empirical results demonstrate that while Pa-Attention introduces marginal computational overhead, our framework maintains training efficiency comparable to standard decoder-only attention. This confirms that the proposed approach incurs negligible latency, remaining well within acceptable complexity bounds.
>
> [1] A Survey on Parallel Reasoning.
>
> [2] See What You Are Told: Visual attention sink in large multimodal models. ICLR2025
>
> [3] Self-Consistency Improves Chain of Thought Reasoning in Language Models.  ICLR2023
>
> [4] Reinforced Attention Learning. Google
>
> We appreciate your valuable feedback. We will incorporate these clarifications into the revised manuscript. We hope our responses have addressed your concerns, and we are happy to provide further clarification if needed.

---

> > ### Author Rebuttal · Reviewer_YoSR · 2026-04-04
> >
> > | Method | Time (s) | Throughput (token/s) |
> > |---|---|---|
> > | Qwen2.5-VL-3B | 266 | 58 |
> > | Sequential | 481 | 53 |
> > | Majority Voting | 1197 | 49 |
> > | Ours (w/o reuse) | 364 | 104 |
> > | Ours (w/ reuse) | 312 | 123 |
> >
> > The result doesn't make sense to me. With majority voting, you can run things in parallel. This result seems to be flawed. I am not convinced with the cost analysis here.

---

> > > ### Author Response · Authors · 2026-04-05
> > >
> > > We appreciate the reviewer’s insightful comment regarding our implementation of Majority Voting. We would like to clarify our design choices and provide further context. First, we acknowledge that our current implementation follows a straightforward serial approach, which aligns with the original Self-Consistency framework[1]. We agree that majority voting can be implemented in various ways, such as parallel sampling across different model instances (full-parallel) or utilizing Top-K sampling within a single forward pass. To address the reviewer's concern, we have conducted a comparative analysis of these implementation strategies, which we summarize in the table below. All the experiments are carried out on V* benchmark. In summary, we would like to emphasize that Majority Voting is an algorithmic framework rather than a single, fixed implementation. Different engineering strategies—such as serial execution versus parallel sampling with prefix caching—result in distinct trade-offs between latency and diversity.
> > >
> > > Majority Voting (Serial): The reasoning efficiency is the slowest.
> > >
> > > Majority Voting (Full-Parallel): Different models are run in parallel, and the inference efficiency is lower than that of Majority Voting (Parallel Top-K), but the effect is better.
> > >
> > > Majority Voting (Parallel Top-K): Single-model employing Top-K sampling. While this strategy enhances inference efficiency by leveraging prefix caching, it may come at the cost of performance.
> > >
> > > | Method | Time (s) | Performance (%) |
> > > | :--- | :---: | :---: |
> > > | Qwen2.5-VL-3B (Baseline) | 266 | 63.3 |
> > > | Sequential | 481 | 65.0 |
> > > | Majority Voting (Serial) | 1197 | 64.7 |
> > > | Majority Voting (Full-Parallel) | 351 | 63.3 |
> > > | Majority Voting (Parallel Top-K) | 329 | 60.4 |
> > > | Ours | 312 | 67.9 |
> > >
> > > As shown in the table, the inference efficiency of full-parallel for Majority Voting is lower than that of Top-K sampling. This is primarily because the former lacks a shared prefix cache, whereas the latter leverages prefix caching to significantly accelerate the inference process. However, the Top-K approach tends to yield lower diversity in reasoning paths, which may result in a slight degradation in overall model performance[1]. Given that our primary objective is to maximize the reasoning capability of the model, we opted for the serial implementation to ensure the highest possible performance.
> > >
> > > We understand that our initial description may have been insufficient, potentially causing confusion regarding our experimental settings. We clarify that our choice of baseline was consistent with standard practices in the field. Crucially, our method consistently outperforms these baselines regardless of the specific engineering implementation chosen. We believe these variations are inherent to the baseline itself and do not affect the validity of our performance claims. Our primary efficiency evaluation remains the comparison against the native Qwen2.5-VL baseline. our efficiency gains were validated through a direct comparison with the native Qwen2.5-VL model. The choice of engineering implementation for Majority Voting does not impact the inference performance of our proposed method.
> > >
> > > We sincerely apologize for the lack of clarity in our initial manuscript. We will include this detailed comparison and the corresponding experimental settings in the revised version of our paper to provide a more comprehensive view of the trade-offs between efficiency and performance.
> > >
> > > Thank you very much for your time and for your careful handling of the review process. We hope that this clarification addresses your concerns. We would be very grateful if you could reconsider your assessment of our work in light of these additional explanations.
> > >
> > >
> > > Reference:
> > >
> > > [1] Self-Consistency Improves Chain of Thought Reasoning in Language Models. ICLR2023

---

### Official Review · Reviewer_zDkc · 2026-03-13

**Soundness:** 3
**Presentation:** 3
**Significance:** 3
**Originality:** 3
**Overall Recommendation:** 4
**Confidence:** 3

**Summary:**

This paper introduces Visual Para-Thinker, a parallel reasoning framework for MLLMs that employs visual partitioning strategies to generate diverse reasoning paths. By integrating Pa-Attention and LPRoPE for path isolation, the model demonstrates significant performance gains across counting, grounding, and hallucination benchmarks. The efficient vLLM-based implementation achieves these improvements with minimal inference overhead.

**Compliance With Llm Reviewing Policy:**

Affirmed.

**Key Questions For Authors:**

see weakness

**Limitations:**

ye s

**Strengths And Weaknesses:**

Strengths

· Presents a novel and intuitive approach to applying parallel thinking in the vision-language domain.

· Technical contributions (Pa-Attention and LPRoPE) are well-motivated and effectively address path interference.
· Extensive experiments show consistent and substantial improvements over strong baselines.

Weaknesses

· Lacks detailed ablation comparing the individual contributions of the two partitioning strategies.

· Does not fully explore how inference time and memory scale with an increasing number of reasoning paths.

---

> ### Author Rebuttal · Authors · 2026-03-28
>
> We appreciate your positive feedback and constructive comments. In the following, we address your key concerns regarding our work:
>
> ### **W1: Lacks detailed ablation comparing the individual contributions of the two partitioning strategies.**
>
> Answer: We appreciate this suggestion. In Section 5 and Fig. 5(a)(b), we analyzed the two strategies, though we acknowledge the presentation could be clearer. The key findings are:
>
> - **Scan-order** excels at counting tasks, where sequential visual scanning is beneficial.
> - **Block-based** is more effective for grounding, hallucination, and visual search tasks, where spatial locality matters.
>
> This complementarity motivates our hybrid training paradigm. Importantly, **at inference time, only one strategy is applied per query** (selected based on task type), so no redundant computation is incurred. This is analogous to Qwen3's thinking/non-thinking training approach. We will make this mechanism more explicit in the revision.
>
> ### **W2:  Does not fully explore how inference time and memory scale with an increasing number of reasoning paths.**
>
> Answer:
> 1. **Inference time.** Table 5 compares the inference time of our method (using 4 paths) on the V* benchmark against several baseline approaches. The results indicate that the inference time of our method is comparable to that of the original Qwen2.5-VL. This negligible overhead is primarily attributable to our efficient engineering optimizations. The table is shown below.
> | Method           | Time (s) $\downarrow$ | Throughput (token/s) $\uparrow$ |
> | :--------------- | :-------------: | :------------------------: |
> | Qwen2.5-VL-3B|266|58|
> | Sequential |481|53|
> | Majority Voting |1197|49|
> | Ours (w/o reuse) |364|104|
> | Ours (w/ reuse) |312|123|
>
> Furthermore, we evaluated the impact of increasing the inference path length on the total inference time based on V* benchmark of our method, as summarized in the table below.
>
> | Method           | Time (s) $\downarrow$ |
> | :--------------- | :-------------------: |
> | Qwen2.5-VL-3B|266|
> | Ours (4 paths) |312|
> | Ours (2 paths) |294|
> | Ours (1 path) |281|
>
> **The results demonstrate that the inference time of our method increases only marginally as the reasoning path length grows. We attribute the marginal increase in inference time to two primary factors:**
>
> **First**, the synchronization overhead inherent in the parallel reasoning stage, where the system must wait for the completion of all paths. However, in the Visual Para-Thinker, the reasoning paths exhibit comparable lengths, thereby minimizing this waiting time.
>
> **Second**, the summarization stage accounts for a significant portion of the total latency. This efficiency is largely due to the high-performance implementation of vLLM, which optimizes parallel inference. Furthermore, the reuse of the KV cache during the summarization stage effectively mitigates the computational overhead by eliminating redundant prefill operations.
>
>
> 2. **Memory.** We provide some additional analysis on the memory usage of Visual Para-Thinker.  During the parallel reasoning stage, the maintenance of KV caches across multiple reasoning paths inevitably increases memory consumption. However, leveraging advanced engineering optimizations—specifically Copy-on-Write, Forking, and KV Cache Reuse—we observed that this overhead remains within 5%, which is well within acceptable limits for practical deployment. Conversely, Test-Time Scaling (TTS) paradigms inherently aim to push the performance boundaries of models by increasing computational and memory overhead. Future work will focus on further optimizing these resource requirements to enhance the efficiency of our approach.
>
> We sincerely thank you once again for your time, effort, and valuable feedback. We will incorporate these clarifications into the revised manuscript. We hope our responses have addressed your concerns, and we are happy to provide further clarification if needed. We truly appreciate your thoughtful review and wish you all the best in your future research endeavors.

---

> > ### Author Rebuttal · Reviewer_zDkc · 2026-04-03
> >
> > Thank you for the effort. I will keep my score.

---

> > > ### Author Response · Authors · 2026-04-05
> > >
> > > Thank you for your response. If all your concerns have been addressed, would you mind considering updating your rating? Your feedback is crucial for the acceptance of our paper, and we would truly appreciate your support in this process.

---

### Decision · Program_Chairs · 2026-04-30

**Decision:**

Accept (regular)

**Comment:**

This paper proposes a parallel reasoning framework for multimodal LLMs. The core idea is pretty intuitive. Instead of going deeper with sequential chain-of-thought, you go wider by running multiple reasoning paths in parallel, each attending to different partitions of the image. The two key components are Pa-Attention for keeping paths isolated from each other, and LPRoPE for making sure paths are positionally unbiased and discriminable. The whole thing is implemented efficiently on top of vLLM with KV cache reuse, and results on counting, grounding, and hallucination benchmarks show consistent improvements. The rebuttal was pretty thorough. The authors added comparisons with new baselines, consistently outperforms all of them. This is the most important result. It directly addresses whether the gains come from the parallel architecture or just from the distilled data, and the answer is clearly the former.